# Neuroplastic Expansion in Deep Reinforcement Learning

**Jiashun Liu**
HKUST

**Johan Obando-Ceron**
Mila - Québec AI Institute
Université de Montréal

**Aaron Courville**
Mila - Québec AI Institute
Université de Montréal

**Ling Pan**[*]
HKUST

## Abstract

The loss of plasticity in learning agents, analogous to the solidification of neural pathways in biological brains, significantly impedes learning and adaptation in reinforcement learning due to its non-stationary nature. To address this fundamental challenge, we propose a novel approach, *Neuroplastic Expansion* (NE), inspired by cortical expansion in cognitive science. NE maintains learnability and adaptability throughout the entire training process by dynamically growing the network from a smaller initial size to its full dimension. Our method is designed with three key components: (*1*) elastic topology generation based on potential gradients, (*2*) dormant neuron pruning to optimize network expressivity, and (*3*) neuron consolidation via experience review to strike a balance in the plasticity-stability dilemma. Extensive experiments demonstrate that NE effectively mitigates plasticity loss and outperforms state-of-the-art methods across various tasks in MuJoCo and DeepMind Control Suite environments. NE enables more adaptive learning in complex, dynamic environments, which represents a crucial step towards transitioning deep reinforcement learning from static, one-time training paradigms to more flexible, continually adapting models. **We make our code publicly available.**

## 1 Introduction

In the realm of neuroscience, it has been observed that biological agents often experience a diminishing ability to adapt over time, analogous to the gradual solidification of neural pathways in the brain (Livingston, 1966). This phenomenon, typically known as the *loss of plasticity* (Mateos-Aparicio & Rodríguez-Moreno, 2019), significantly affects an agent's capacity to learn continually, especially when agents learn by trial and error in deep reinforcement learning (deep RL) due to the non-stationarity nature. The declining adaptability throughout the learning process can severely hinder the agent's ability to effectively learn and respond to complex or non-stationary scenarios (Abbas et al., 2023). This limitation presents a fundamental obstacle to achieving sustained learning and adaptability in artificial agents, which echoes the *plasticity-stability dilemma* (Abraham & Robins, 2005) observed in biological neural networks.

There have been several recent studies highlighting a significant loss of plasticity in deep RL (Kumar et al., 2021; Lyle et al., 2022), which substantially restricts the agent's ability to learn from subsequent experiences (Lyle et al., 2023; Ma et al., 2024). The identification of primacy bias (Nikishin et al., 2022) further illustrates how agents may become overfitted to early experiences, which inhibits learning from subsequent new data. The consequences of plasticity loss further impede deep RL in continual learning scenarios, where the agent struggles to sequentially learn across a series of different tasks (Dohare et al., 2024).

Research on addressing plasticity loss in deep RL is still in its early stages, with recent approaches including parameter resetting (Nikishin et al., 2022; Sokar et al., 2023) (or its advancement with random head copies (Nikishin et al., 2024)), and several implementation-level techniques like normalization, activation functions, weight clipping, and batch size adjustments (Obando Ceron et al., 2023; Nauman et al., 2024; Elsayed et al., 2024). However, reset-based methods often lead to performance

---

[*]Corresponding author, email: lingpan@ust.hk

instability and training inefficiency due to the need for period re-training, while implementation-level modifications lack generalizability and do not directly target the plasticity issue. The field currently lacks a unified methodology that can effectively address plasticity loss while maintaining training stability and efficiency across varying environments.

Humans adapt to environmental changes and novel experiences through cortical cortex expansion in cognitive science (Welker, 1990; Hill et al., 2010). This process involves the gradual activation of additional neurons and the formation of new connections to facilitate the ability to learn continually. Drawing inspiration from this biological mechanism, we propose a novel perspective – *Neuroplastic Expansion*, which can help maintain plasticity in deep RL. The key insight is that an agent starting learning with a smaller network and dynamically growing to a larger size (ultimately reaching the original static network dimension) can effectively tackle plasticity loss by maintaining a high level of elastic neurons throughout training (Figure 1). *Elastic neurons means neurons that have plasticity. In this paper, the words elastic and plastic are interchangeably*. We first provide empirical evidence in (§3.1) validating its potential for mitigating plasticity loss and improving final performance in certain cases, even with a naive incremental expansion. We provide a detailed description of plasticity loss and pruning for RL in Appendix A.

Building upon this insight, we systematically introduce *Neuroplastic Expansion* (NE), a simple yet effective mechanism to maximize the benefits of incremental growth training (§3.2). NE adds high-quality elastic candidates based on potential gradients (Evci et al., 2020). However, this will lead to increased computation costs for always enlarging the network, and cannot fully leverage its expressivity as elastic neurons, which are activated and can be updated to fit the new data, may turn dormant during the course of training. As visualized in Appendix F.6, dormant neurons can diminish representational capacity and lead to suboptimal behavior.

To address this challenge, NE introduces a reactivation process for dormant neurons, pruning them and potentially reintroducing them as candidates in subsequent growth stages; thereby better utilizing the network expressivity and enhancing the policy's sustainable learning ability.

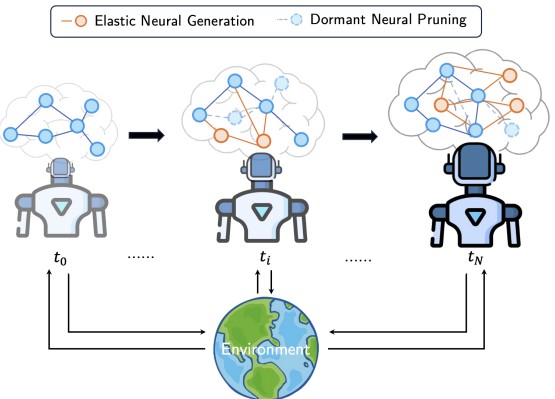

Figure 1: **High-level illustration of Neuroplastic Expansion RL**. The network regenerates elastic neurons based on gradient potential, recycles dormant neurons, and undergoes progressive topology growth to mitigate plasticity loss. The agent consolidates neurons through experience review, preserving prior helpful knowledge and ensuring policy stability.

Unlike previous reset-based approaches (such as resetting the last layers, introducing copies of heads or reinitializing parts of the network) that risk forgetting and require periodic re-training, NE maintains learning continuity in a smoother way. To further mitigate potential instability from continuous topology adjustments, we introduce an effective *Experience Review* technique that consolidates neurons by adaptively reviewing prior knowledge during the later training stages, ensuring policy stability and balancing the stability-plasticity dilemma. Extensive experiments on long-term training, continual adaptation, and vision RL tasks demonstrate the effectiveness of our method across various scenarios, showcasing its ability to maintain plasticity while ensuring policy stability.

The main contributions are summarized as follows:

- We introduce a novel mechanism, *Neuroplastic Expansion* (NE), to mitigate the loss of plasticity in deep RL.

- We develop effective activated neuron generation and dormant neuron pruning mechanism for better network capacity utilization, and a neuron consolidation technique for preventing forgetting helpful reusable knowledge.

- We conduct extensive experiments in standard RL tasks, which demonstrate the effectiveness of NE across various tasks including MuJoCo (Todorov et al., 2012) and DeepMind Control Suite (DMC) (Tassa et al., 2018a) tasks that outperform previous strong baselines by a large margin, and can be effectively adapted to continual learning scenarios.

## 2 BACKGROUND

**Deep Reinforcement Learning** The reinforcement learning problem can be typically formulated by a Markov decision process (MDP) represented as a tuple $(\mathcal{S}, \mathcal{A}, \mathcal{P}, \mathcal{R}, \gamma)$, with $\mathcal{S}$ denoting the state space, $\mathcal{A}$ the action space, $\mathcal{P}$ the transition dynamics: $\mathcal{S} \times \mathcal{A} \times \mathcal{S} \rightarrow [0, 1]$, $\mathcal{R}$ the reward function: $\mathcal{S} \times \mathcal{A} \rightarrow \mathbb{R}$, and $\gamma \in [0, 1)$ the discount factor. The agent interacts with the unknown environment with its policy $\pi$, which is a mapping from states to actions, and aims to learn an optimal policy that maximizes the expected discounted long-term reward. The state-action value of $s$ and $a$ under policy $\pi$ is defined as $Q^\pi(s, a) = \mathbb{E}_\pi[\sum_{t=0}^{\infty} \gamma^t \mathcal{R}(s_t, a_t)|s_0 = s, a_0 = a]$.

In actor-critic methods, the actor $\pi_\phi$ and the critic $Q_\theta$ are represented using neural networks as function approximators with parameters $\phi$ and $\theta$ (Fujimoto et al., 2018; Haarnoja et al., 2018). The critic network is updated by minimizing the temporal difference loss, i.e., $\mathcal{L}_Q(\theta) = \mathbb{E}_D\big[(Q_\theta(s, a) - Q_\theta^\mathcal{T}(s, a))^2\big]$, where $Q^\mathcal{T}(s, a)$ denotes the bootstrapping target $\mathcal{R}(s, a) + \gamma Q_{\bar\theta}(s', \pi_{\bar\phi}(s'))$ computed using target network parameterized by $\bar\phi$ and $\bar\theta$ based on data sampled from a replay buffer $D$. The actor network $\phi$ is typically updated to maximize the Q-function approximation according to $\nabla_\phi J(\phi) = \mathbb{E}_D\left[\nabla_a Q_\theta(s, a)|_{a=\pi_\phi(s)} \nabla_\phi \pi_\phi(s)\right]$.

**Activated Neuron Ratio** Activated neurons are easily updated by new data. Therefore, the proportion of activated neurons in the network, i.e. activated neuron ratio is often considered positively correlated with plasticity Ma et al. (2024). It is defined as the proportion of neurons whose output exceeds a certain threshold $\tau$ in the neural network, effectively counting all neurons except dormant ones (Xu et al., 2024). Formally, a neuron $i$ in layer $l$ is considered activated when output $h_i(x) > 0$ (in contrast to dormant neurons (Sokar et al., 2023)):

$$f(l_i) = \frac{\sum_{i \in l, x \in Id} 1(h_i(x) > 0)}{N}, \tag{1}$$

where $Id$ denotes the input distribution. The relationship between the Activated Neuron Ratio curve and plasticity is as follows: The primary cause behind an agent's plasticity loss is the rapid dormancy of numerous neurons as training progresses, which diminishes the model's representational capacity (Sokar et al., 2023; Qin et al.). Consequently, delaying the reduction in activated neurons is commonly considered as a way to mitigate plasticity loss, i.e, slowing the downward trend of the curve. The Activated Neuron Ratio is widely used for plasticity visualization (Ma et al., 2024) due to its intuitive interpretability.

## 3 NEUROPLASTIC EXPANSION RL

In this section, we begin by discussing the insights of Neuroplastic Expansion RL empirically, and analyze its effect on mitigating the loss of plasticity in Section 3.1. Then, we systematically present the details of our method in Section 3.2.

### 3.1 ILLUSTRATION OF INSIGHTS OF DYNAMIC GROWING RL

There have been several recent studies investigating loss of plasticity in supervised learning (Lyle et al., 2023; Lewandowski et al., 2024), which corresponds to the phenomenon where neural networks gradually lose the ability to adapt and learn from new experiences (Lin et al., 2022). This is further exacerbated in RL (Sokar et al., 2023; Nikishin et al., 2024), due to the non-stationary nature and the tendency to overfit prior knowledge, resulting in suboptimal policies. A predominant approach to tackle plasticity loss is based on resetting (Nikishin et al., 2022), where the agent resets the last few layers of its neural network periodically throughout training. However, although effective, this kind of approach typically forgets helpful and reusable knowledge which is critical for learning, and thus the agent experiences a drop in performance once resetting is applied.

A major cause for agents to adapt to novel circumstances is the growth of the human cortex (Hill et al., 2010). Motivated by this insight, we propose a novel perspective that RL agents can maintain high plasticity through dynamic neural expansion mechanisms, sustaining the policy's continual adaptation ability based on new experiences. Initially, data collected by a random policy may be of lower quality and therefore require less network expressivity to fit (Burda et al., 2019; Zhelo et al., 2018). As the agent improves, it tends to require a more expressive network for fitting the more complex value estimates and policies. Maintaining neuronal vitality and regeneration during training can significantly mitigate plasticity loss, as it allows for ongoing adaptation to new information.

Motivated by this insight, we propose a novel angle, Neuroplastic Expansion, where the agent starts with a smaller network that gradually evolves into a larger one throughout the learning process. We first present a naive implementation of this idea and then systematically discuss our methodology in Section 3.2.

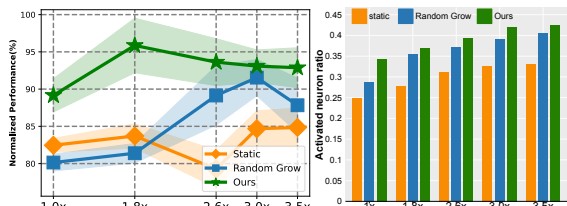

Concretely, we adopt a small-to-large neural expansion approach, where the initial network capacity is 20% of the size of a typical full TD3 (Fujimoto et al., 2018) network (i.e., uniformly selected 20% of neurons from each layer of the full TD3 architecture). To validate the potential of

Figure 2: **Comparison of normalized performance (left) and the ratio of active neurons (right) for vanilla TD3** (Fujimoto et al., 2018) and its variants: with a naive random growth network and our NE method, across varying network capacities.

this novel perspective, we first consider the simplest approach for growing the network by uniformly adding $k$ new connections to the current critic topology $\breve{\theta} \subset \theta$ and the current actor topology $\breve{\phi} \subset \phi$ for each layer $l \in N$, which then participate in gradient propagation according to $\mathbb{I}_{grow} = \mathrm{Random}_{i \notin \breve{\theta}_l}(\theta_l, k) \cup \mathrm{Random}_{i \notin \breve{\phi}_l}(\phi_l, k)$.

We build this idea upon TD3 and conduct fair comparisons with vanilla TD3 (i.e., with static actor and critic networks) using different network capacities. We compare the performance of our dynamically growing network against a naive topology growth strategy and static networks in standard TD3 (Fujimoto et al., 2018) using the HalfCheetah environment. To ensure fairness in comparison, the final network size of NE is set to be identical to that of the other methods. Therefore, the comparison with the vanilla TD3 algorithm is fair given the same network capacity at the end of training.

From the left side of Figure 2, we observe that even a naive incremental network growth strategy leads to noticeable performance improvements, particularly as network capacity increases. The right side of Figure 2 demonstrates the ratio of active neurons (Sokar et al., 2023) for each algorithm during training. This metric evaluates whether RL agents maintain their expressive capacity, which is positively correlated with plasticity retention. As shown, topology growth effectively reduces neuron deactivation, thereby preserving the agent's ability to learn policies, mitigating plasticity loss, and alleviating primacy bias.

## 3.2 METHOD: NEUROPLASTIC EXPANSION

Hill et al. (2010) demonstrated that the cerebral cortex expands in response to environmental stimuli, learning, and novel experiences. This dynamic growth and reorganization enhance cognitive and decision-making abilities, fostering adaptability to changing situations. Motivated by our preliminary results and inspired by this biological mechanism, we present *Neuroplastic Expansion* (NE), a training framework for RL agents that employs a dynamically expanding network to retain learning capacity and adaptability throughout the entire learning process.

**Elastic Topology Generation**  While incremental random topology growth has contributed to maintaining plasticity, we aim to elucidate the underlying mechanisms that drive its effectiveness. To this end, we explore the dormant neuron theory (Sokar et al., 2023), a prevalent explanation for plasticity loss. This theory posits that neurons within a network become inactive during training, typically when their activation values approach zero (Lu et al., 2019). When this occurs, these neurons lose their ability to learn, reducing the network's overall capacity to efficiently process

new situations. Their outputs gradually diminish, ultimately falling below the activation function threshold (Lu et al., 2019). Meanwhile, prior research has shown that parameters that lack prompt exposure to high-gradient stimuli are more prone to dormancy. Drawing on these findings, we derive the following observation:

> *Naive random topology growth sporadically introduces new connections that can generate significant gradients for neurons nearing dormancy, thereby supplying them with strong backpropagation signals and allowing them to continue learning.*

If the above conjecture holds in deep RL, then selecting topology candidates based on gradient magnitude may mitigate plasticity loss more effectively than randomly expanding network connections. To this end, we follow the sparse network training framework in Tan et al. (2022). Before each topology expansion, we sample a batch of transitions from buffer $D$ to compute the gradient magnitude across all model parameters, for-

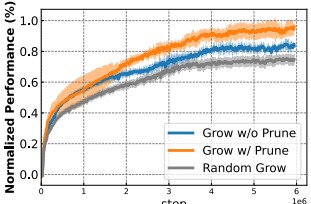 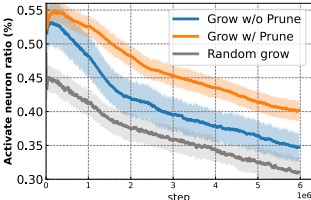

Figure 3: **Performance comparison and plasticity evaluation of random growth**, our proposed growth method, and pruning mechanisms. All experiments were conducted with seven independent seeds.

malized as $|\nabla_\theta L|$. Then, we select $k$ connections that yield the highest gradient magnitudes to augment the topology: $\mathbb{I}_{grow} = ArgTopk_{i \notin \theta^l}(|\nabla_\theta^l L|)$. The topology growing process $G_k$ adaptively starts every $\Delta T$ step. We try several growing schedules to achieve warm topology growth to prevent training instability caused by significant network modifications. We chose cosine annealing through practice (see Appendix F.5 for experiments): at timestep $t$ the $k$ is decayed in a cosine annealing manner: $\frac{\alpha}{2}(1 + \cos(\frac{t\pi}{T_{end}}))$, where $\alpha$ denotes the discount factor and $T_{end}$ is the shut down step. Finally, we restrict the weights of the newly included neurons (which are randomly initialized) to a specific range for training stabilization (Nauman et al., 2024) via weight clipping (Elsayed et al., 2024). Following standard practice (Liu et al., 2022), we start applying weight clipping on both the input and output weights of a neuron once it is incorporated into the topology.

The results in Figure 3 indicate that substituting random growth with gradient-based selection of new connections enhances agent performance and mitigates dormant neuron issues. Additionally, we compare two commonly used topology initialization strategies: uniform and Erdos-Rrenyi (Evci et al., 2020), and we find that initializing the network using the Erdos-Rrenyi type is slightly more effective for RL. This is because Erdos-Rrenyi enables the number of connections in a sparse layer to scale with the sum of the input and output channels, leading to more stable computations in smaller networks. Consequently, the agent can achieve more efficient learning in the early stages of training.

**Dormant Neuron Pruning**   After replacing the naive growth strategy with our gradient-guided growth process $G_k$, a new challenge emerges, i.e., dormant neurons experience a significant decline in representational potency while continuing to occupy network capacity. These inactive neurons may be inherent from initialization or introduced into the network topology through the growth schedule. This situation can lead to a topology that saturates quickly, resulting in disproportionately low computational power. Moreover, Lu et al. (2019); Liu et al. (2019) demonstrated that parameters exclusively forward-linked to dormant neurons are omitted from the gradient calculation process. As a result, they become ineffective, as they fail to receive meaningful guidance signals. We hypothesize that pruning and resetting dormant neurons following topology expansion can address this issue.

This process aims to free up space for new connections, allowing dormant neurons to be reactivated as new candidates. According to Ceron et al. (2024a;b); Sokar et al. (2025), sparsifying the value-based agents' network can often enhance performance, which empirically supports our real-time pruning approach. Consequently, we introduce a synchronous prune-and-reset mechanism into the topology growth schedule $G_k$. Here, we adopt the calculation method $f(\cdot)$ shown in Eq. 1 and set $\tau$ to be 0, which means only considering fully dormant neurons: $\mathbb{I}_{prune} = \{\text{index}(\theta_i)|f(\theta_i) = 0\}$. To facilitate the topology's gradual growth and avoid over-pruning, we employ the *Truncate process* (that operates on the pruning set $\mathbb{I}_{prune}^l$) which drops excess elements randomly from the set when

its size exceeds a predefined upper bound. The new truncated set is then used for pruning (*Refer to Appendix D.2 for the detailed pseudo-code of our Truncate Process*). In practice, we define the pruning upper bound in each layer $l$ as $\omega \times |\mathbb{I}_{grow}^l|$, where $|\mathbb{I}_{grow}^l|$ represents the number of elements in the current grow set. The parameter $\omega \in [0, 1)$ acts as a "discount" factor, moderating the pruning relative to the total network growth in layer $l$. Thus, the two final parameters controlling topology evolution are $k$ and $\omega$. We denote our grow-prune schedule as $G_{k,\omega}$. The complete growth-pruning schedule for each layer $l$ in actor $\phi$ and critic $\theta$ networks is defined as:

$$G_{k,\omega} : \begin{cases} 1.\ \mathbb{I}_{grow}^l = ArgTopk_{i \notin \breve{\phi}^l}(|\nabla_\phi^l L_t^\phi|) \cup ArgTopk_{i \notin \breve{\theta}^l}(|\nabla_\theta^l L_t^\theta|) \\ 2.\ \mathbb{I}_{prune}^l = \left\{ \text{Index}(\breve{\phi}_i^l) | f(\breve{\phi}_i^l) = 0 \right\} \cup \left\{ \text{Index}(\breve{\theta}_i^l) | f(\breve{\theta}_i^l) = 0 \right\}; \end{cases}. \tag{2}$$

We then use $\mathbb{I}_{grow}^l, \mathbb{I}_{prune}^l$ to generate $\{0, 1\}$, which are dot-multiplied with the network parameters to achieve the evolution of the actor topology $\breve{\phi}$ and critic topology $\breve{\theta}$. The results in Figure 3 demonstrate that introducing the pruning mechanism after each growth round further preserves the plasticity of the RL network and enhances overall performance. NE is inspired by plasticity injection (PI) (Nikishin et al., 2024). While both NE and PI maintain plasticity by extending networks, PI focuses on one-time large-scale network growth, whereas our method achieves continuous real-time adaptation.

Notably, our sparse network training framework for maintaining stable learning is summarized as follows: (i) Inspired by Evci et al. (2020), we use Erdos-Rrenyi initialization to ensure that the number of connections in a sparse layer scales with the sum of the output and input channels, thereby enhancing initial stability. (ii) We employ cosine annealing to guide gradual network growth, reducing noise caused by network changes. (iii) Following sparse training practices in RL (Tan et al., 2022), we keep the first and last layers dense to ensure the stability of the encoding and decoding.

**Neuron Consolidation via Experience Review** As discussed above, the proposed elastic topology generation and dormant neuron pruning-based dynamic growth approach effectively mitigate plasticity loss. This method maintains a higher proportion of active neurons, leading to improved performance. However, a closer examination of the results, i.e. Tab. 8, reveals that NE demonstrates a high standard deviation across seeds in the later stage of training, suggesting a potential risk of performance oscillation near convergence. We hypothesize that frequent network topology changes may introduce slight probabilistic errors in the network structure. Consequently, as the activated neuron ratio decreases, some agents may risk forgetting previously learned skills from the early training stages (e.g., standing in the HalfCheetah task). This phenomenon leads to suboptimal behavior post-convergence and increases variance between seeds.

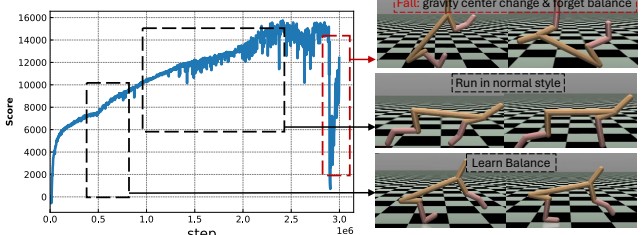

Figure 4: Instability of NE in the later stage without neuron consolidation via experience review.

We examined the behaviors on the HalfCheetah task and found that there are multiple runs in which the agent no longer could stand after a fall (Figure 4). This observation strengthens our confidence in the above conjecture. To address this, we seek a topology-agnostic and simple mechanism to further enhance the stability of NE by mitigating catastrophic forgetting. Meanwhile, prior studies (Rolnick et al., 2019; Zhou et al., 2020) show that experience replay (ER) helps mitigate forgetting and supports network plasticity in both RL and supervised learning. Inspired by these findings, we aim to develop an ER mechanism sensitive to both the training stage and the activated neuron ratio.

In practice, we define the absolute slope of the activated neuron ratio curve as: $\Delta(f(\theta)) := \sum_{i \in \theta} |\Delta f(\theta_i)|$ as the threshold $\epsilon$. Specifically, $\epsilon$ measures the slope of changes in the number of activated neurons (within the critic network) during the recent $c$ steps, i.e., $[150, 450]$. A larger positive value of $\epsilon$ corresponds to a greater reduction in dormant neurons, indicating active plasticity improvements in our dynamically growing network. In contrast, a small $\epsilon$ (close to zero) suggests a

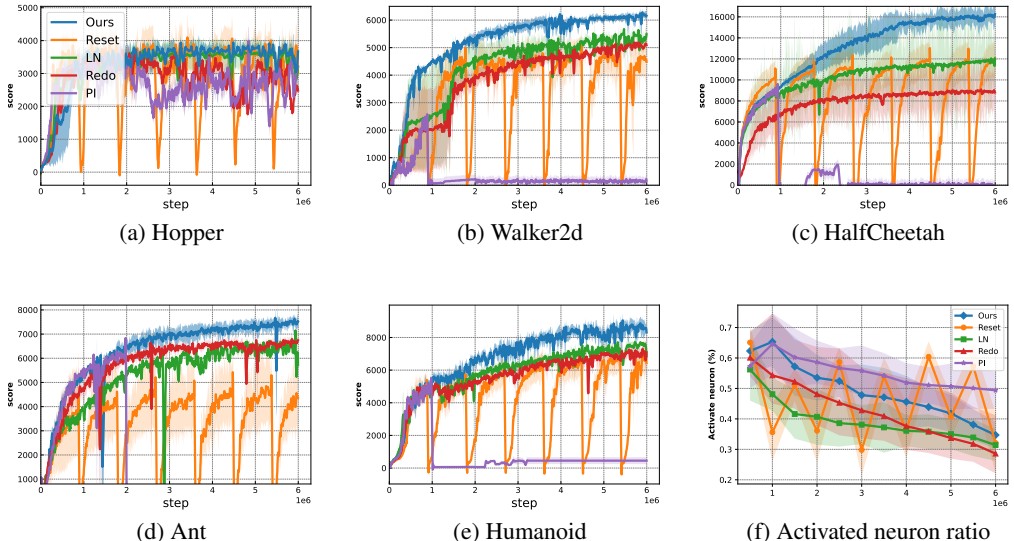

Figure 6: **Performance comparisons on OpenAI MuJoCo environments (Todorov et al., 2012):** (a) Hopper, (b) Walker2D, (c) HalfCheetah, (d) Ant, (e) Humanoid. (f) Results of the activated neuron ratio. NE outperforms the baselines in four tasks and maintains a high activated neuron ratio.

bottleneck in plasticity gains, meaning limited capacity for further topology expansion and marking the later stage of policy learning (see Figure 5).

During training, we observe that $\epsilon$ generally decreases over time, since elastic neuron generation and dormant neuron pruning significantly impact the number of dormant neurons in the early training stage. However, due to limited network capacity, they cannot indefinitely increase the proportion of active neurons. To counteract this effect, we encourage the agent to review earlier experiences from the buffer when dormant neurons are more prevalent. At each training step, a random number $m \in [0,1]$ is selected, if $m > \epsilon$, the sampling is conducted from the initial quarter of the buffer; otherwise, the conventional sampling method is employed. We provide pseudocode outlining the full algorithmic process in Appendix D.1.

## 4 EXPERIMENTS

In this section, we conduct comprehensive experiments to evaluate whether NE can alleviate the loss of plasticity. We investigate the following key questions: (**i** (§4.1)) In a long-term training setting, can NE effectively combat early data overfitting, enhance performance, and attenuate the decline in activation neuron ratio? (**ii** (§4.2)) In continuous adaptation task, can NE enable agents to efficiently adapt to new tasks while slowing the dormancy rate of activated neurons, without being constrained by prior learning limitations?

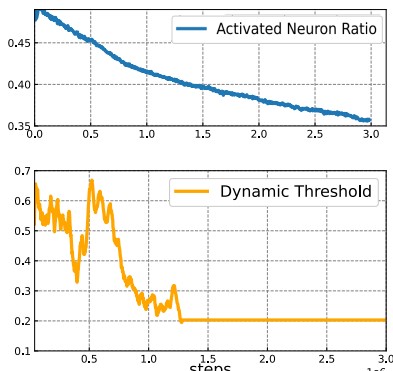

Figure 5: How $\epsilon$ changes with plasticity measurement. In practice, we set a lower bound for $\epsilon$.

(**iii** (§4.3)) What are the effects of NE on the policy and value networks, and what is the importance of its different components? (**iv** (§4.4)) Can our method be applied to other deep RL methods with more complex image inputs?

### 4.1 LONG-TERM TRAINING TASKS

**Experimental Setup** We conduct a series of experiments based on the standard continuous control tasks from OpenAI Gym (Brockman, 2016) simulated by MuJoCo (Todorov et al., 2012) with long-term training setting, i.e. 3M steps → 6M. We compare NE against strong baselines includ-

ing Reset (Nikishin et al., 2022), ReDo (Sokar et al., 2023), Layer Normalization (LN) (Lyle et al., 2024b), and Plasticity Injection (PI) (Nikishin et al., 2024). To ensure a fair comparison and mitigate implementation bias, we base all baselines on official implementations (Sokar et al., 2023; Nikishin et al., 2024; 2022), with the same architecture (See Appendix C.1). All baselines are implemented using the TD3 (Fujimoto et al., 2018) algorithm (where results based on other backbone deep RL methods are discussed in Section 4.4). To ensure a fair comparison and reproducibility of the expected performance of all baselines, we follow the recommended hyperparameter setting from Elsayed et al. (2024) (see Appendix C) for all baselines. The clipping parameter $\kappa$ is set to 3. Each algorithm is trained with 7 random seeds, and we report the mean and standard deviation. A detailed description of hyperparameters and experimental setup is in Appendix C.2. For all tasks, NE initializes agents at 25% of total capacity and grows asymptotically. The growth termination scale is aligned with baselines. The score denotes undiscounted episodic return.

**Results**    As shown in Figure 6, the Reset method suffers from periodic performance drops due to resetting a large portion of layer parameters. This leads to the loss of reusable knowledge, requiring relearning, which impedes overall progress. LN and ReDo demonstrate more stable performance than Reset, but result in a lower proportion of active neurons. PI maintains the highest level of neuron activation, but performs well only in the relatively simpler Hopper environment. NE achieves significant and consistent improvements in learning efficiency and final performance, with a larger margin in more complex environments. Furthermore, NE effectively mitigates plasticity loss, maintaining higher active neuron ratios throughout training compared to the most competitive LN method. Additionally, NE achieves a superior trade-off between performance and neuron utilization. It performs more stably and outperforms PI, which, despite achieving the highest neuron activation, exhibits suboptimal performance in most tasks.

## 4.2    CONTINUAL ADAPTATION

In this section, we investigate the continual ability to learn and adapt to changing environments, which is an essential capability of deep RL agents (Willi* et al., 2024; Elsayed & Mahmood, 2024).

**Experimental Setup**    We follow the experimental paradigm of Abbas et al. (2023) and evaluate our method on a variant of MuJoCo tasks that introduce non-stationarity due to changing environments over time. Specifically, the agent is trained to master a sequence of 4 environments (HalfCheetah → Humanoid → Ant →Hopper), starting with HalfCheetah for 1000 episodes of training, followed by the next task for the same number of episodes. A cycle is completed when training on Hopper is finished. The agent repeats this sequence three times, forming a long-term training schedule. Each task maintains an output head (1 layer MLP), which is trained in conjunction with the backbone network. Our method is applied to all layers except the output head. We compare our approach against the vanilla TD3 agent and the most competitive baseline, Reset.

**Results**    The performance of each method in a single environment, when repeatedly learning across a sequence of environments as described above, is summarized in Figure 8. We aim to evaluate (**i**) the efficiency of the policy in quickly learning a new task after mastering the previous one within a single cycle; (**ii**) the agent's ability to retain the benefits of initial learning for the same task across multiple cycles. As shown, Resetting, which involves initializing the parameters after each task, is currently considered the most effective approach.

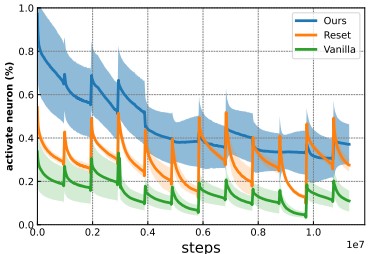

Figure 7:    Activated neuron ratio, recorded for three agents with different settings during cyclic training.

According to results in Figure 8, NE effectively handles each task without being negatively influenced by prior training. It significantly outperforms the original TD3, reinforcing that standard RL algorithms suffer from catastrophic forgetting in a continual learning setting. We attribute this success to NE's ability to introduce real-time topology adjustments, which facilitate storing new knowledge and ensuring continuous learning capabilities.

Notably, in certain environments, such as Ant, NE-TD3 retains prior knowledge and seamlessly continues learning. This could be attributed to the experience review module and the preservation of old neurons within the topology, which store critical information essential for successful continual learning. When analyzing column-wise comparisons, NE performs comparably to Reset in most tasks, yet it significantly outperforms Reset in the challenging Humanoid task. The comparison of active neurons for each baseline is shown in Figure 7, which illustrates that even during extended training, NE effectively mitigates neuron dormancy while maintaining efficient learning. In fact, it even outperforms Reset in the early stages of training, despite Reset reinitializing all parameters.

The spike phenomenon in Figure 7 is related to research examining the relationship between input distribution and plasticity. Lu et al. (2019) observed that neurons failing to activate due to the activation function can be reactivated when the input distribution to the network changes. A similar phenomenon has been reported in Ma et al. (2024), where data augmentation was found to alleviate the decline in activated neurons. In the continuous adaptation task we tested, a sudden change in the environment may have altered the input data distribution, leading to a temporary increase in the activated neuron rate. However, after a short training period, the neurons returned to a dormant state.

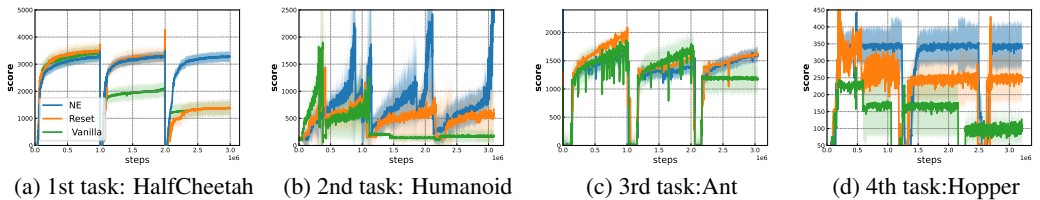

(a) 1st task: HalfCheetah    (b) 2nd task: Humanoid    (c) 3rd task:Ant    (d) 4th task:Hopper

Figure 8: Learning curves of three methods (*NE, Reset, Vanilla*) on continuous adaptation tasks.

## 4.3 ABLATION STUDY

In this section, we conduct an additional ablation study to investigate the effectiveness of neuron consolidation in Neuroplastic Expansion, while the importance of elastic neural generation and dormant neuron pruning has already been demonstrated in Figure 3. Additionally, we examine the experience review (ER) module and assess the impact of our method on the policy and value networks.

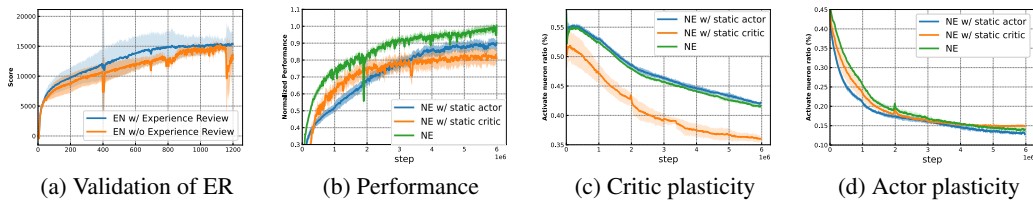

(a) Validation of ER    (b) Performance    (c) Critic plasticity    (d) Actor plasticity

Figure 9: Ablation study: (a) presents the validity verification of ER, while (b) to (d) illustrate the impact of NE on the actor and critic from the perspectives of plasticity and performance.

As illustrated in Figure 9a, the results on HalfCheetah show ER improves the training stability, particularly in the later stage. Additionally, we select 4 tasks with image input and sparse reward setting to further verify the effectiveness of ER. The results in Appendix F.4 indicate that when assisted with ER, our training framework consistently stabilizes the convergent policy at a high level across different seeds (i.e., lower standard deviation). In contrast, the absence of this critical component makes the model more vulnerable to policy collapse in later stages (albeit to a lesser extend than Reset). This demonstrates the importance of reviewing historical knowledge in maintaining policy stability and preventing the loss of early valuable information, as discussed in Section 3.2.

Figure 9b illustrates a comparative analysis of our method and its variants, isolating the effects of NE on the actor and critic networks separately. We observe that NE plays a particularly crucial role in the critic network. Removing NE from the critic results in significant performance degradation and substantial plasticity loss compared to the full version of NE (Figure 9c, 9d). In contrast, limiting expansion to only the critic leads to a less pronounced drop in performance. This asymmetry in impact may be attributed to the greater challenge faced by the critic, as it must provide accurate value

estimates for various state-action pairs (Meng et al., 2021) under a non-stationary data distribution. In addition, the critic provides guidance signals for actor optimization, which requires a high level of adaptability (Fujimoto et al., 2024).

## 4.4 VERSATILITY

In this section, we demonstrate the generality of our proposed method by applying it to another popular deep RL algorithm and evaluating its performance on more complex tasks with image inputs.

**Experimental Setup** We evaluate four image-based motion control tasks from the DeepMind Control Suite (DMC) (Tassa et al., 2018b): Reacher Hard, Reacher Easy, Walker Walk, and Cartpole Swingup Sparse, to demonstrate the generalization of NE across deep RL algorithms and its robustness across diverse tasks. For the backbone algorithm, we use DrQ (Yarats et al., 2022), a well-known variant of SAC (Haarnoja et al., 2018) specifically designed for processing image inputs. We compare our approach against the same baselines used in Section 4.1. The weight clipping parameter $\kappa$ is set to 2.

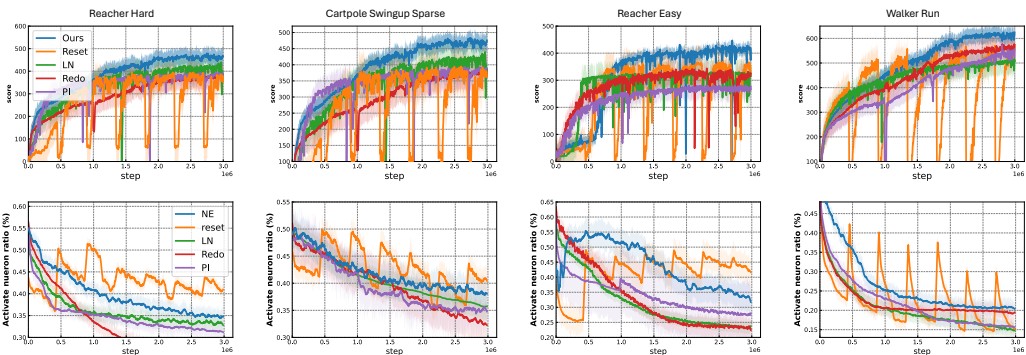

Figure 10: The first row shows the performance of five methods. The second row corresponds to the percentage of activated neurons. All experiments were run with seven independent seeds.

**Results** The first row of Figure 10 shows that NE can be effectively combined with DrQ to achieve stable performance on image-input tasks, outperforming all baselines and demonstrating its versatility across different algorithms. Additionally, the second row of Figure 10 (activated neuron ratio) indicates that although NE does not reach the peak activation levels of Reset, it delays neuron dormancy more effectively than other warm mitigation strategies, ensuring stable performance.

## 5 CONCLUSION

Inspired by the cortical growth in biological agents that triggers neuronal topology expansion to enhance plasticity and adaptability to new situations, we introduce a novel perspective for deep RL agents to mitigate the loss of plasticity through a dynamic growing network. Building on this insight, we present a comprehensive topology growth framework, *Neuroplastic Expansion* (NE), which addresses both growth and pruning mechanisms to maximize the performance benefits of topology expansion. NE adds high-quality elastic neurons and connections based on the gradient signals to improve the network's learning ability; prunes dormant neurons in a timely manner for better utilizing network capacity, and returns them to the candidate set to maintain the agent's plasticity, making dormant neurons reusable. NE outperforms previous methods that aimed at alleviating plasticity loss on diverse tasks and demonstrates stable, outstanding performance on plasticity metrics. Future directions include exploring adaptive architectures that dynamically adjust their capacity based on the agent's performance or task complexity, as well as further reducing computational overhead. Overall, despite its acknowledged limitations, NE provides practical insights into enhancing the plasticity of deep RL agents. We hope our findings pave the way for future research, potentially leading to more sample-efficient and adaptable deep RL algorithms.

ACKNOWLEDGMENT

This work is supported by the National Natural Science Foundation of China 62406266. We would also like to thank the Python community (Van Rossum & Drake Jr, 1995; Oliphant, 2007) for developing tools that enabled this work, including NumPy (Harris et al., 2020) and Matplotlib (Hunter, 2007).

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

## A    RELATED WORK

**Plasticity loss in RL**    Recent studies have found that agents guided by the non-stationary objective characteristic of RL suffer catastrophic degradation (Lyle et al., 2023) and ultimately overfit early experiences, which limits their continuous learning ability. This phenomenon is known as *plasticity loss*. Behaviorally, the loss of plasticity during training can manifest as, for instance, an inability to obtain an effective gradient to guide policy adaptation in fine-tuning (Dohare et al., 2024), limiting potential transferability and making it difficult to adapt to new sampling and unseen data (refer to as *primacy bias* (Nikishin et al., 2022)). While the research on this topic is still in its early stages, several techniques have been demonstrated to mitigate the loss of plasticity in deep RL. Resetting global networks or dormant neurons  (Nikishin et al., 2022; Sokar et al., 2023) and modifying activation functions (Abbas et al., 2023) are observed to mitigate plasticity loss. Lyle et al. (2024b) empirically finds that network normalization, particularly layer regularization (LN), can maintain plasticity. Nikishin et al. (2024) introduced copies of random heads for injecting plasticity to improve continual learning ability. PLASTIC (Lee et al., 2023) divides plasticity into input plasticity and label plasticity, addressing them separately. NaP (Lyle et al., 2024a) alleviates the over-fitting issue from the perspective of the association between regularization and learning rate. In pixel-based deep RL, data augmentation (Ma et al., 2024) and batch size reduction (Obando Ceron et al., 2023) have been analyzed for their effectiveness in reducing plasticity loss. Recent studies have found that mitigating gradient starvation (Dohare et al., 2024) and constraining deviations from initial weights (Kumar et al., 2023; Lewandowski et al., 2024) can also be effective in mitigating plasticity loss. In supervised learning, Lee et al. (2024a) proposed to use a combination of fast and slow update networks to balance between fast, fleeting adaptation and slow, steady generalization in supervised learning. Ash & Adams (2020); Berariu et al. (2021) found that both warm initialize and pre-train can better adapt to increasing amounts of data.

**Pruning in RL**    Sokar et al. (2021) showed that training the deep RL policy with a changing topology is difficult due to training instability.  Since then, this challenging topic has been well-studied. Policy Pruning and Shrinking (PoPs) (Livne & Cohen, 2020) obtain a sparse deep RL agent with iterative parameter pruning. Graesser et al. (2022); Tan et al. (2022); Lee et al. (2024b) attempt to train a sparse neural network from scratch without pre-training a dense teacher. Existing methods mainly focus on obtaining a smaller topology to improve training efficiency.  However, our paper explores whether growing the topology from small to large can mitigate the plasticity loss in deep RL agents rather than training with the maximum number of parameters. In deep RL, the model size commonly utilized is typically small and may encounter policy collapse after scaling up (Schwarzer et al., 2023; Ceron et al., 2024a). Thus, sparse training techniques that have proven beneficial for significantly reducing training costs in fully supervised domains may not yield similar advantages in deep RL in terms of computation cost reduction. Instead, we provide a new perspective and focus on making deep RL agents better use of the growing network size to further improve their learning ability.

## B    RELATED PRELIMINARIES

**TD3**    In our paper, we use TD3 as a practice backbone for most experiments.Thus, here we introduce the training process of TD3 in detail. TD3 is a deterministic Actor-Critic algorithm. Different from the traditional policy gradient method DDPG, TD3 utilizes two heterogeneous critic networks,i.e., $Q_{\theta_{1,2}}$, to mitigate the over-optimize problem in Q learning.

The training process of TD3 follows the Temporal difference learning (TD):

$$L_Q(\theta_i) = \mathbb{E}_{s,a,s'}\big[(y - Q_{\theta_i}(s,a))^2\big] \text{ for } \forall i \in \{1,2\}. \tag{3}$$

Where $y = r + \gamma \min_{j=1,2} Q_{\bar{\theta}_j}(s', \pi_{\bar{\phi}}(s'))$, $\bar{\phi}$ denotes the target network parameters.  The actor is updated according to the Deterministic Policy Gradient (Fujimoto et al., 2018) At each training step $t$, the agent first randomly samples a batch of state transitions from the buffer and calculates the loss gradient as described above.  It is worth noting that the proposed method in our paper is a plug-in and does not affect the training process of the reinforcement learning algorithm itself.

**DrQ** DrQ is an algorithm based on maximum entropy reinforcement learning. Its training method are the same as SAC (Haarnoja et al., 2018). The soft value function is trained to minimize the squared residual error:

$$J_V(\phi) = \mathbb{E}_s \frac{1}{2}(V_\phi(s) - Q_\theta(s, a) + \log \pi_\psi(a|s)) \tag{4}$$

where $a$ is sampled according to the current policy instead of the replay buffer. The soft Q-function parameters can be trained to minimize the soft Bellman residual: $J_Q(\theta) = \mathbb{E}_{s,a} \sim D[\frac{1}{2}(Q_\theta(s, a) - \hat{Q}(s, a))^2]$. Then the policy network can be learned by directly minimizing the expected KL divergence

$$J(\psi) = \mathbb{E}_{s \sim D}[D_{KL}(\pi_\phi(\cdot|s) || \frac{\exp Q(s)}{Z(s)})] \tag{5}$$

## C EXPERIMENTAL DETAILS

### C.1 NETWORK STRUCTURE

**TD3** The normal size of the TD3 network we used is the official architecture, and the detailed setting is shown in Tab 1. Besides, the network we used for Scale-up experiments incrementally adds the depth of both the actor and critic. For the $1.8\times$ agent, we add one MLP as a hidden layer in two critic networks in front of the output head and one MLP layer as the embedding layer at the front of the actor. And the hidden dim is also increased to 256. And so on, we follow the above approach to grow the model by an equal amount each time.

| Layer | Actor Network | Critic Network |
|---|---|---|
| Fully Connected | (state dim, 256) | (state dim, 256) |
| Activation | ReLU | ReLU |
| Fully Connected | (256, 128) | (256, 128) |
| Activation | ReLU | ReLU |
| Fully Connected | (128, action dim) and (128, 1) | |
| Activation | Tanh | None |

Table 1: Network Structures for TD3

**DrQ** We use DrQ to conduct experiments on robot control tasks within DeepMind Control using image input as the observation. All experiments are based on previously superior DrQ algorithms and maintain the architecture from the official setting unchanged.

### C.2 IMPLEMENTATION DETAILS

**TD3** Our codes are implemented with Python 3.8 and Torch 1.12.1. All experiments were run on NVIDIA GeForce GTX 3090 GPUs. Each single training trial ranges from 6 hours to 21 hours, depending on the algorithms and environments, e.g. DrQ spends more time than TD3 to handle the image input and DMC needs more time than OpenAI mujoco for rendering. Our TD3 is implemented with reference to `github.com/sfujim/TD3` (TD3 source-code). The hyper-parameters for TD3 are presented in Table 2. Notably, for all OpenAI mujoco experiments, we use the raw state and reward from the environment and no normalization or scaling is used. An exploration noise sampled from $N(0, 0.1)$ (Lillicrap, 2015) is added to all baseline methods when selecting an action. The discounted factor is $0.95, 0.99$ and we use Adam Optimizer (Kingma, 2014) for all algorithms. Tab.2 shows the common hyperparameters of TD3 used in all our experiments. Following (Elsayed et al., 2024), We start applying weight clipping on both input and output weights of a neuron once it becomes part of the topology. To ensure a fair comparison and reproducibility of the expected performance of all baselines, we employ the recommended hyperparameter setting in Elsayed et al. (2024) for all baselines, with the clipping parameter $\kappa = 3$ for MuJoCo tasks and $\kappa = 2$ for image-input tasks. In our method, dormant neurons are randomly reinitialized after pruning, preparing them to rejoin the network topology. When they are re-selected, weight clipping is applied to the

corresponding neurons. Weight clipping is a well-established, straightforward, and generic implementation technique (that operates independently of core algorithm design and specific algorithm choices, rather than a new algorithmic contribution here), which is a standard practice that we apply consistently across all implemented methods. Weight clipping is activated after the network initialization and operates throughout the training process. Additionally, we introduce layer normalization after the hidden layers of both the critic and actor networks to establish the LN-TD3 baseline for comparison. Notably, other methods do not incorporate normalization mechanisms to ensure a fair comparison. We reproduce Plasticity Injection Following the guidelines from the official paper (Nikishin et al., 2024). We utilize the default settings of the official ReDo codebase for our comparative analysis https://github.com/google/dopamine/tree/master/dopamine/labs/redo. For Reset-TD3, after a lot of empirical debugging, we found that parameter reset every $90,000$ step is a good setting, and fixed it as a hyperparameter. We suggest using seed 5 to reproduce the learning curve in Figure 5.

| Hyperparameter | NE-TD3 | Reset-TD3 | ReDo-TD3 | LN-TD3 | PI-TD3 | |
|---|---|---|---|---|---|---|
| Actor Learning Rate | $1e^{-4}$ | $1e^{-4}$ | $1e^{-4}$ | $1e^{-4}$ | $1e^{-4}$ | $1e^{-4}$ |
| Critic Learning Rate | $1e^{-3}$ | $1e^{-3}$ | $1e^{-3}$ | $3e^{-4}$ | $3e^{-4}$ | $1e^{-3}$ |
| Representation Model Learning Rate | None | None | None | $1e^{-4}$ | $5e^{-3}$ | $5e^{-3}$ |
| Discount Factor | 0.99 | 0.99 | 0.99 | 0.99 | 0.99 | 0.99 |
| Batch Size | 128 | 128 | 128 | 128 | 128 | 128 |
| Buffer Size | 1e6 | 1e6 | 1e6 | 1e6 | 1e6 | 1e6 |

Table 2: A comparison of common hyperparameter choices of algorithms. We use 'None' to denote the 'not applicable' situation.

**DrQ** We use DrQ as the backbone to verify our methods on DeepMind Control tasks. And All the methods use the same setting. The details can be seen in Tab.3. Notably, For Reset-DrQ, we only reset the last three MLP layers which is suggested by their paper.

| Hyperparameter | NE-DrQ | Reset-DrQ | ReDo-DrQ | LN-DrQ | PI-DrQ | |
|---|---|---|---|---|---|---|
| Replay buffer capacity | 1e6 | 1e6 | 1e6 | 1e6 | 1e6 | 1e6 |
| Action repeat | 2 | 2 | 2 | 2 | 2 | 2 |
| Seed frames | 4000 | 4000 | 4000 | 4000 | 4000 | 4000 |
| Exploration steps | 2000 | 2000 | 2000 | 2000 | 2000 | 2000 |
| n-step returns | 3 | 3 | 3 | 3 | 3 | 3 |
| Mini-batch size | 256 | 256 | 256 | 256 | 256 | 256 |
| Discount $\gamma$ | 0.99 | 0.99 | 0.99 | 0.99 | 0.99 | 0.99 |
| Optimizer | Adam | Adam | Adam | Adam | Adam | Adam |
| Learning rate | $1e-4$ | $1e-4$ | $1e-4$ | $1e-4$ | $1e-4$ | $1e-4$ |
| Agent update frequency | 2 | 2 | 2 | 2 | 2 | 2 |
| Critic Q-function soft-update rate | 0.01 | 0.01 | 0.01 | 0.01 | 0.01 | 0.01 |
| Features dim. | 50 | 50 | 50 | 50 | 50 | 50 |
| Hidden dim. | 1024 | 1024 | 1024 | 1024 | 1024 | 1024 |

Table 3: A default set of hyper-parameters used in our DrQ based methods.

**NE** For Humanoid and Ant tasks, we set grow interval $\Delta T = 25000$, grow number $k = 0.01 *$ rest capacity, Prune upper bond $\omega = 0.4$, ending step is the max training step, the threshold of ER is 0.35 and the decay weight $\alpha = 0.02$(which is used in all the tasks). For other OpenAI Mujoco tasks, we set grow interval $\Delta T = 20000$, grow number $k = 0.15 *$ rest capacity, Prune upper bond $\omega = 0.2$, ending step is the max training step, the threshold of ER is 0.25. Notably, as for the continuous adaptation tasks, we set $\Delta T = 50000$ to ensure the topology can continue growing during the long-term training, and other parameters are same as Humanoid. When we combine our method with DrQ, We use single set of hyperparameters to apply on all four tasks. $\Delta T$ is set as 20000 to ensure the network can grow quickly to handle the complex input. The grow number $k$ is $0.02 *$ rest capacity for critic and $0.01 *$ rest capacity for actor. The Prune upper bond $\omega = 0.23$ and the threshold of ER is 0.18. We set ER step $C$ to 450 for Mujoco tasks and $\{150, 250\}$ is work on DMC. Additional optimizer resets were eliminated for all methods to ensure the fairness of the experiment.

# D    PSEUDOCODE CODE

## D.1    PSEUDO-CODE FOR NE

---

**Algorithm 1** Neuroplastic Expansion TD3

---

$\pi_\phi$: All parameters in actor. $Q_{\theta_{\{1,2\}}}$: All parameters in critics, $M_l$: Sparse mask in layer $l$. Initial sparse rate $Sp$. Set parameter $\kappa$ and the clip bounds for all layers: $\{s_l^\phi = \frac{1}{\sqrt{n_l^\phi}}\}_{l=1}^L$, $\{s_l^\theta = \frac{1}{\sqrt{n_l^\theta}}\}_{l=1}^L$ for weight clipping operation, where $\sqrt{n_l}$ denotes the number of the neurons in layer $l$.

\# Initialize the sparse networks for stable starting
keep $\theta^{l \in \{1,N\}}, \phi^{l \in \{1,N\}}$ dense;   $\theta, \phi \leftarrow \text{Erdos-Renyi}(Sp)$

---

**Neuroplastic Expention (every $\Delta T$)**

\# Calculate growing number at $t$ step to achieve warm growing
$k \leftarrow cosine\ annealing(t, T_{end})$
**for** each $l_\phi \in \pi_\phi, l_\theta \in Q_{\theta_{\{1,2\}}}$ **do**
  \# Select top $k$ weights from candidates
  $\mathbb{I}_{grow} = \text{ArgTop}k_{i^l \notin \breve\phi^l}(|\nabla_\phi^l L_t^\phi|) \cup \text{ArgTop}k_{i^l \notin \breve\theta^l}(|\nabla_\theta^l L_t^\theta|)$
  \# Collect the weights related to selected dormant neurons
  $\mathbb{I}_{prune}^l = \{\text{Index}(\breve\phi_i^l)|f(\breve\phi_i^l) = 0\} \cup \{\text{Index}(\breve\theta_i^l)|f(\breve\theta_i^l) = 0\}$
  Start *truncate process*                                          ▷ Algorithm 2
  Get indexes from $\mathbb{I}_{grow}, \mathbb{I}_{prune}$
  Generate topology mask map $M_{l_\phi}, M_{l_\theta}$
  \# Update new topology
  $\breve\theta_l \leftarrow \theta_l \odot M_{l_\theta}, \breve\phi_l \leftarrow \phi_l \odot M_{l_\phi}$
  \# Weight clipping for the newly added neurons
  $\theta_{\mathbb{I}_{grow}^l} \leftarrow \text{Weight Clipping}(\theta_{\mathbb{I}_{grow}^l}, \min = -\kappa s_l^\theta, \max = \kappa s_l^\theta)$
  $\phi_{\mathbb{I}_{grow}^l} \leftarrow \text{Weight Clipping}(\phi_{\mathbb{I}_{grow}^l}, \min = -\kappa s_l^\phi, \max = \kappa s_l^\phi)$
**end for**

---

**Train the RL policy**

$a \leftarrow \pi_{\breve\phi}(s)$ (with Gaussian noise)
Observe $r$ and new state $s'$
Fill $D$ with $(s, a, r, s')$
\# Experience review
**if** $random(0, 1) > \Delta f(\theta)$ **then**
  sample a batch from bottom $\frac{1}{4}D$
**else**
  sample a batch from total $D$
**end if**
Update $Q_{\breve\theta_{\{1,2\}}}, \pi_{\breve\phi}$ based on TD3

---

## D.2 Pseudo-code for Truncate Process

---
**Algorithm 2** Truncate Process

---
1: **Input:**
2: # Enter the necessary inputs
3:    $min\ value$: min pruning threshold (set to be 0)
4:    $\mathbb{I}_{prune}^{l}$: the pruning set
5:    $\mathbb{I}_{grow}^{l}$: the growth set # for determining max pruning amount
6:    $\omega$: a preset ratio in $[0, 1)$ # for determining max pruning amount
7: **Output:** Truncated pruning set
8:
9: $max\ value = \omega \times |\mathbb{I}_{grow}^{l}|$ # max pruning amount. $|\cdot|$ represents the number of the elements in the set
10: **if** $|\mathbb{I}_{prune}^{l}| > max\ value$ **then**
11:    Randomly remove excess elements from the pruning set $\mathbb{I}_{prune}^{l}$
12: **end if**
13: **Return** $\mathbb{I}_{prune}^{l}$

---

# E DETAILED EXPLANATION OF THE GROWTH MECHANISM

We follow the classical sparse network exploration method in (Evci et al., 2020). We use all the weights to calculate the gradient backpropagation because we want to find a weight subgraph composed of multiple candidates to add to the current network, that is, the newly added weight subsets jointly obtain high gradient magnitude under the influence of each other, and calculating the weights one by one will ignore the interaction between them and other weights in the subset, resulting in serious errors and high-cost computation. To this end, we compute the full weight to cheaply select the valid subgraphs. However, this evaluation does introduce errors that add low-quality weights, and the reason why this growth framework is effective is that real-time pruning removes the error caused by growth. In the future, we will further optimize the growth mechanism to improve the accuracy of its evaluation.

# F ADDITIONAL EXPERIMENTS

## F.1 ANALYSIS OF GROW & PRUNE RATE

We conducted ablation experiments on dormant neuron pruning across two hard tasks in Tab.4. The findings demonstrate the significant and generalized benefits of dormant neuron pruning for our framework.

| Method | Cartpole Swingup Sparse | Walker Walk |
|--------|-------------------------|-------------|
| w/ prune | $425.09 \pm 45.28$ | $491.28 \pm 40.62$ |
| w/o prune | $382.14.28 \pm 29.86$ | $367.51.24 \pm 43.06$ |

Table 4: Performance (Average of 10 runs).

## F.2 ANALYSIS OF GROW & RRUNE RATE

In table.5, we set experiments for growth rate and pruning rate analyses. Our method shows good performance on both image-based and state-based tasks with a very slow growth schedule, i.e. $0.001 \sim 0.009$, but it is not sensitive to pruning rate, i.e., keeping it between $10\%$ and $40\%$

| Task | 0.0005 | 0.00010 | 0.00050 | 0.00100 | 0.00150 |
|------|--------|---------|---------|---------|---------|
| state-input (Ant) | $7529.15 \pm 132.36$ | $7749.25 \pm 127.73$ | $7714.28 \pm 147.56$ | $7308.47 \pm 261.95$ | $7287.39 \pm 384.02$ |
| image-input (DMC Reacher Hard) | $416.72 \pm 50.37$ | $442.31 \pm 56.42$ | $475.46 \pm 34.11$ | $431.74 \pm 52.09$ | $419.38 \pm 37.24$ |

Table 5: The evaluations of growth rate on vector-input and image-input tasks. Average of 3 runs.

## F.3 ANALYSIS OF STARTING SPARSITY RATE

We empirically find that starting from around 0.25 is a general and effective setting for growth. For the cycle adaptation task we find that starting from 0.15 is a good choice. In the future, we will explore how to find the appropriate sparsity adaptively according to the task.

We uniformly sample different starting sparsity rates and test the performance. In table.6, we find that the effect is similar when the sparsity ratio is between $[0.7, 0.8]$, too sparse network cannot learn efficiently in the early stage of training, and too dense network will limit the gain brought by network growth. Therefore, we set it to 0.75 uniformly.

| Task | 0.85 | 0.8 | 0.75 | 0.7 | 0.65 |
|---|---|---|---|---|---|
| state-input (Ant) | $6962.52 \pm 297.41$ | $7694.37 \pm 115.28$ | $7749.25 \pm 127.73$ | $7681.61 \pm 108.07$ | $7255.34 \pm 212.86$ |
| image-input (DMC Reacher Hard) | $425.95 \pm 27.62$ | $480.73 \pm 35.18$ | $475.46 \pm 34.11$ | $492.13 \pm 22.53$ | $432.67 \pm 29.17$ |

Table 6: The evaluations of initial sparsity on vector-input and image-input tasks. Average of 3 runs.

## F.4 VERIFY THE EFFICIENCY OF EXPERIENCE REPLAY MODULE

The experience replay module is related to reducing forgetting in the later training stage and enhancing stability-plasticity ability. The experiment depicted in Fig.7a illustrates that frequent alterations to the network topology can induce sudden shifts in network computations, which may prune neurons holding valuable information, leading to instability in subsequent training phases. The trajectory visualization (Figure 12) demonstrates that the agent may exhibit suboptimal behavior by forgetting the initial skill (standing up). Consequently, we introduce the experience replay mechanism during the later stages of training to prompt the network to revisit earlier data, mitigating forgetting and fostering stability in the later training phases.

To further authenticate the efficacy of ER across a spectrum of tasks, we focus on four challenging image input tasks: Reacher Hard and Walker Walk in DMC, along with two sparse reward manipulator control tasks: Hammer (sparse) and Sweep Into (sparse). The results in Table.7 shows enhanced performance and notably reduced variance following the integration of the experience replay mechanism. This outcome underscores that experience replay fosters training stability.

| Method | Reacher Hard | Walker Walk | Hammer sparse (success rate) | Sweep Into sparse (success rate) |
|---|---|---|---|---|
| NE w/ ER | $475.46 \pm 34.11$ | $491.28 \pm 40.62$ | $0.54 \pm 0.13$ | $0.61 \pm 0.09$ |
| NE w/o ER | $419.25 \pm 65.73$ | $427.31 \pm 73.84$ | $0.47 \pm 0.21$ | $0.52 \pm 0.16$ |

Table 7: The evaluations of initial sparsity on vector-input and image-input tasks. Average of 3 runs.

In addition, we set a study of experience review (ER) on various tasks in Tab.8-10, and the results show that without ER, although all the agents with different seeds can learn efficiently in the early stage, the performance will drop at the later training stage and become unstable in some runs(high std). We also find that this phenomenon is more pronounced on complex sparse reward tasks as well as image input scenarios. Therefore we suggest employing ER in complex tasks to assist our method for stable training.

| Method | at $1/2$ training stage | at $3/4$ training stage | end of training |
|---|---|---|---|
| NE w/ ER | $6635.12 \pm 245.28$ | $7258.18 \pm 151.66$ | $7749.25 \pm 127.73$ |
| NE w/o ER | $6673.54 \pm 216.41$ | $6896.92 \pm 264.23$ | $7235.14 \pm 493.83$ |

Table 8: Performance on vector-input task (Ant) (Average of 10 seeds).

## F.5 ANALYSIS OF GROW MODE

The choice of the control mode for topology growth is empirical, and we have tested several modes in the experimental phase, i.e uniform schedule, warm decay, and cosine annealing. We show the comparison in Table 11, the results show that although all modes are efficient, cosine annealing performs the best. We guess that this is because it includes cyclicality based on warm decay, which is more robust in practice.

| Method | at $1/2$ training stage | at $3/4$ training stage | end of training |
|---|---|---|---|
| NE w/ ER | $408.22 \pm 74.14$ | $462.09 \pm 48.74$ | $475.46 \pm 34.11$ |
| NE w/o ER | $413.75 \pm 79.22$ | $439.02 \pm 86.15$ | $419.25 \pm 65.73$ |

Table 9: Performance on image-input task (Reacher Hard) (Average of 10 seeds).

| Method | at $1/2$ training stage | at $3/4$ training stage | end of training |
|---|---|---|---|
| NE w/ ER | $396.69 \pm 83.27$ | $416.39 \pm 55.26$ | $424.76 \pm 47.68$ |
| NE w/o ER | $397.15 \pm 79.22$ | $421.02 \pm 61.08$ | $402.88 \pm 71.23$ |

Table 10: Performance on sparse reward task (Cartpole Swingup Sparse)(Average of 10 seeds).

| Mode | Cartpole Swingup Sparse |
|---|---|
| Cosine annealing | $425.09 \pm 45.28$ |
| Uniform | $389.66 \pm 41.53$ |
| warm decay | $392.24 \pm 31.95$ |

Table 11: The evaluations of growth mode on vector-input and image-input tasks. Average of 5 runs.

## F.6 VISUALIZATION OF THE NETWORKS WITH LOW ACTIVATED NEURON RATIO

When the activated neurons ratio is low, means the networks contain lots of dormant (dead) neurons that retain useless (negative) weights. In this section, we add two visualizations to demonstrate the detrimental effects of accumulating lots of dormant neurons in a network: it diminishes the network's overall representational capacity during training and leads to suboptimal policy.

We contrasted the backpropagation gradient heatmaps of a standard network with those of a network containing numerous dormant neurons (Figure 11), where darker colors indicate higher gradients and faster weight optimization. Both networks use the same data for input. The findings indicate that when a network has a substantial number of dormant neurons, the gradient guidance is notably diminished, thereby constraining the learning efficacy. This limitation stems from dormant neurons being unable to trigger the activation function, rendering them unseen on the computational graph and impeding the backflow of gradients.

We compare the behaviors of the regular policy and the policy featuring numerous dormant neurons in Figure 12, it becomes evident that the latter tends to exhibit suboptimal behavior due to its reduced learning capacity.

## F.7 ABLATION STUDY OF REPLAY RATIO

We test the effect of different Replay Ratio (RR) on all four DMC tasks. The results in Figure 13 show that with the increase of RR, the performance of baseline and NE increased slightly (the effect of our method was more obvious). We concluded that this was because changing RR proved to be an effective way to alleviate plasticity loss in VRL (Ma et al., 2024).

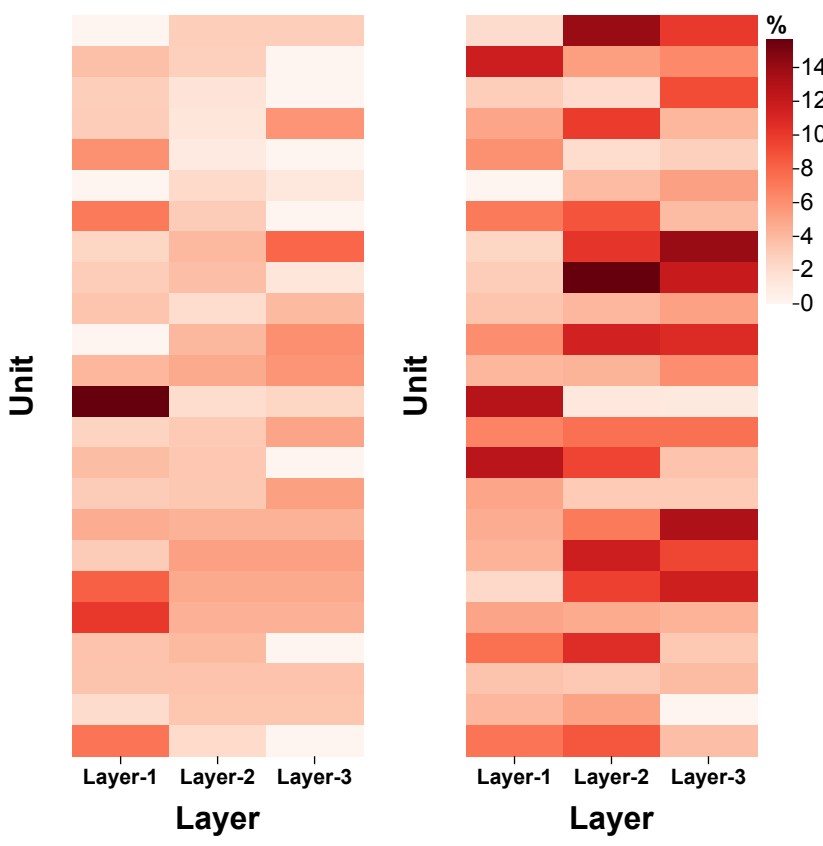

Figure 11: Heat map. Darker colors indicate higher gradients and faster weight optimization. Both networks use the same data for input.

**Dormant neuron**: Although moving, fail to learn the desired running skills

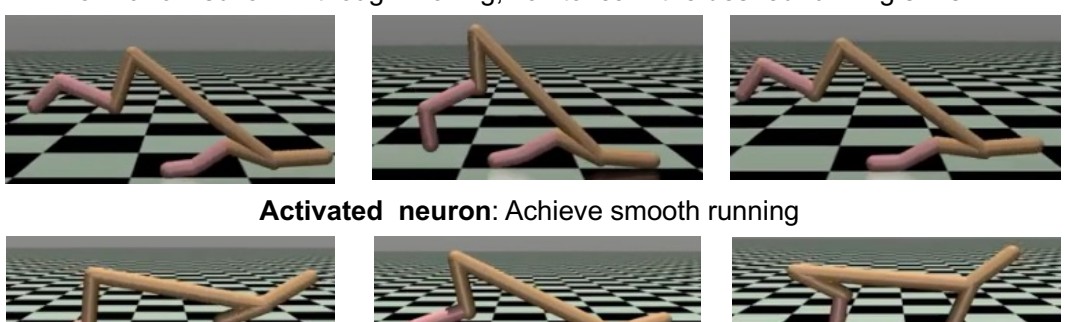

**Activated neuron**: Achieve smooth running

Behavior after Policy training

Figure 12: Comparison of behaviors after training.

## F.8 DETAIL RESULTS OF DIFFERENT NETWORK SIZE

Figure 14 shows the results for uniform fine-grained size sampling.

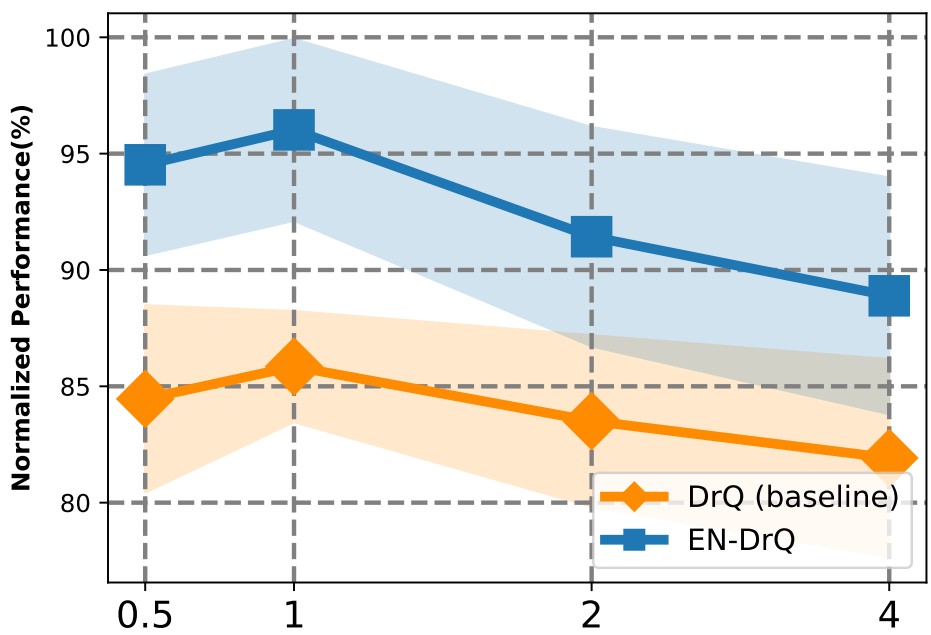

Figure 13: Analysis of different Replay Ratios values.

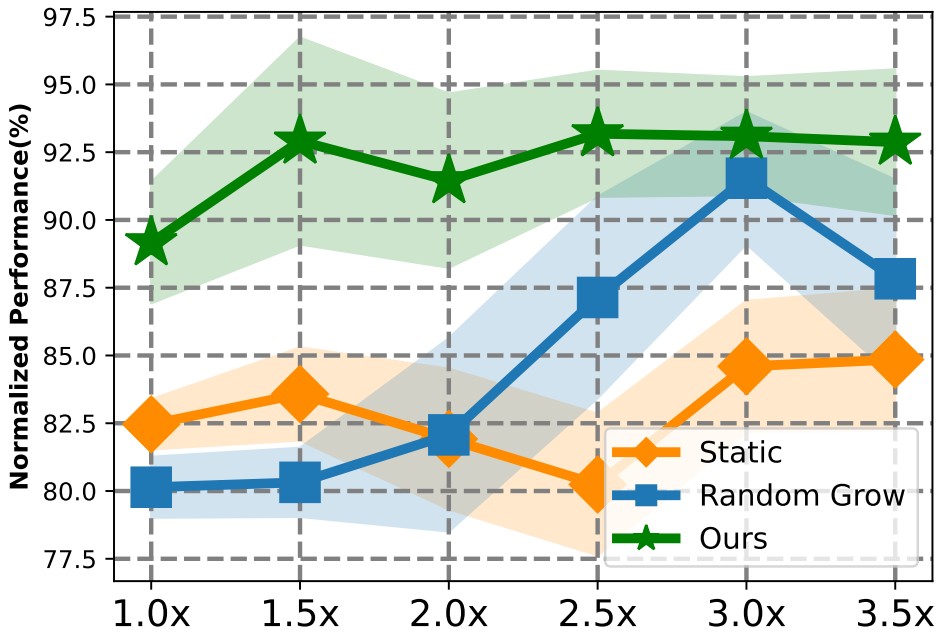

Figure 14: Performance comparison for different final model sizes.

