# OpenReview forum: "Neuroplastic Expansion in Deep Reinforcement Learning"
_ICLR.cc/2025/Conference — ICLR 2025 Poster_

### Official Review · Reviewer_46hD · 2024-10-28

**Soundness:** 3
**Presentation:** 2
**Contribution:** 3
**Rating:** 6
**Confidence:** 5

**Summary:**

This paper addresses the critical issue of plasticity loss in deep reinforcement learning (deep RL), where agents' adaptability decreases over time, hindering continuous learning in dynamic environments. Inspired by biological neural networks, the authors propose Neuroplastic Expansion (NE), a novel mechanism that dynamically enlarges the neural network by adding elastic neurons based on gradient potential. NE maintains high plasticity by regenerating and recycling dormant neurons, effectively mitigating the plasticity-stability dilemma inherent in deep RL.

**Strengths:**

* **Novelty:** The paper introduces a novel idea inspired by human brain mechanisms, specifically cortical expansion, to address plasticity loss in deep RL. This biologically motivated approach offers a novel perspective for significant advancements in continual learning for artificial agents.
* **Good architectural design:** Neuroplastic Expansion (NE) is meticulously designed to balance network growth with resource efficiency. By adding elastic neurons based on gradient potential and recycling dormant neurons, NE maintains network expressivity and adaptability without causing uncontrolled growth in network size.

**Weaknesses:**

* **Missing Relevant Work:**
    * Plasticity-Related Studies: The paper overlooks several relevant studies on neural network plasticity ([1]-[5]).
    * Cortical Expansion Citations: The discussion on cortical cortex expansion cites works that describe patterns of cortical expansion. I think the authors miss foundational studies that first identified evidence of cortical expansion.
* **Experimental Setup**:
    * The evaluation of MuJoCo environments is limited to the TD3 algorithm, which is considered outdated. Assessing NE using more recent and robust algorithms such as TD7, TD-MPC2, or BRO would enhance the relevance and robustness of the findings.
    * Other than Mujoco, I think it is beneficial to test in the state-based DMC, maybe by trying to compare with reset-based methods under identical experimental configurations as primacy bias paper.

[1] On warm-starting neural network training., Ash et al, 2020.

[2] A study on the plasticity of neural networks., Berariu et al, 2021.

[3] PLASTIC: Improving Input and Label Plasticity for Sample Efficient Reinforcement Learning., Lee et al, 2023.

[4] Slow and Steady Wins the Race: Maintaining Plasticity with Hare and Tortoise Networks., Lee et al, 2024.

[5] Normalization and effective learning rates in reinforcement learning., Lyle et al, 2024.

**Questions:**

* How is the network reinitialized based on the growth criterion? Is it initialized with random weights similar to the initial stage?
* If reinitialization involves random weights, how does this approach effectively reduce dormant neurons, especially considering that large feature values after extended training might lead to immediate pruning of reinitialized neurons.
* Is the Experience Review technique effective across all experimental setups, or is its efficacy primarily validated only in specific environments like HalfCheetah?
* PThe performance curves indicate that the dynamic actor maintains a stable plasticity rate similar to the static one. Why does the dynamic actor perform better despite this similarity in plasticity rates?
* For a more comprehensive evaluation, should the authors include additional baselines such as the base TD3 and TD3 with only network growth (without pruning) to isolate the effects of different components of NE.
* What constitutes a valid starting sparsity rate for NE?
* What are the optimal rates for growth and pruning, and how do these rates influence overall performance? An analysis of hyperparameter sensitivity would provide deeper insights into NE's robustness and adaptability.

---

### Official Review · Reviewer_df5D · 2024-10-29

**Soundness:** 2
**Presentation:** 1
**Contribution:** 2
**Rating:** 6
**Confidence:** 4

**Summary:**

This paper presents Neuroplastic Expansion (NE), a generally applicable training scheme designed to mitigate plasticity loss in RL. NE comprises of three components:

1. Adding neuron connections based on potential gradients (elastic neuron generation)
2. Pruning neuron connections based on dormant ratio (dormant neuron pruning)
3. A training batch sampling scheme that focuses on early samples depending on dormant ratio fluctuation (neuron consolidation)

Compared to prior methods such as Reset, ReDo and Plasticity Injection, NE showed superior performance in state-based Mujoco tasks (with TD3) and several pixel-based DMC tasks (with DrQ). NE was also able to maintain plasticity while sequentially training through multiple environments in a cyclic manner. Its plasticity — measured by dormant ratio — is well preserved in the majority of the experiments, proving its effectiveness in maintaining trainability and preventing loss of plasticity.

**Strengths:**

1. The core idea of NE seems very promising in terms of lifelong learning: add new capacity to learn new information, remove useless/dead neurons, and prevent catastrophic forgetting. Connection to biology is also a big plus.
2. The necessity for each component was well explained (Figure 2,3,4). I found it especially interesting to see a proof of catastrophic forgetting in a Mujoco task.

**Weaknesses:**

1. The writing is sometimes not detailed enough and causes confusion (see questions and weaknesses below).
2. Some crucial design choices are not well justified and/or validated.
    1. Neuron consolidation is proposed to prevent catastrophic forgetting, which often occurs late stage (as shown in Figure 4). However, the dynamic threshold they use to control the strength of consolidation plateaus to its lowest value (strongest consolidation) even before halfway through the training process (Figure 5). This discrepancy raises question on whether this complexity is really necessary, especially since a simple time-dependent scheduling scheme could also fit the justification of ‘not forgetting early state-action distribution’.
    2. The amount of pruned dormant neurons are forced to be less than amount of added connections in order to guarantee that the network is increased in size. This also looks like an unnecessary detail since we can achieve the same by using ReDo and then growing a small number of neurons. I think there should be an explanation on why this design choice is essential.
    3. Although less critical, some components of elastic neuron generation also needs more careful consideration, such as the cosine annealing schedule (especially since RigL [1] was not primarily designed for continual learning).
3. The experimental setup needs more refinement.
    - For the main experiments, although it’s convincing that NE surpasses prior works in Mujoco and DMC tasks, it would be nice to see whether NE is also effective in more challenging environments.
    - The recently proposed NaP [2] is an extremely competitive method for continual learning, and I think it’s a crucial baseline in the main experiment.
    - The cycling Mujoco experiment (Figure 9) is plotted on ‘episodes’ (and is the only one). This is problematic, since different lengths of episodes would result in different number of update steps and thus varying degree of plasticity loss.
    - It would have been nice to see whether NE can synergize with other methods such as CReLU or LayerNorm.

Overall, I think this paper needs more improvements before it can be published. However, I am also ready to change my scores if some of the above concerns are refuted/addressed.

[1] Rigging the Lottery: Making All Tickets Winners., Evci et al., ICML 2020.

[2] Normalization and effective learning rates in reinforcement learning., Lyle et al., arXiv.

**Questions:**

1. In Figure 5, at which point does the model reach its maximum capacity (i.e., no more room for growing unless pruned)?
2. In Figure 6, why does Plasticity Injection fail even before injecting in Walker2d and HalfCheetah? Before injection, shouldn’t they be equivalent to vanilla TD3?
3. In the 'Dormant Neuron Pruning' section, the expression $Clip(a,b,c)$ is confusing to read without any definition.
4. The dynamic threshold defined in paragraph in 'Neuron Consolidation' as a whole is too confusing to read. I don't think $\nabla f(\theta)$ is the right definition, since it's not a derivative w.r.t. the dormant ratio, but rather an average change rate of the dormant neuron count. Another thing I want to make sure is that  $\nabla f(\theta)$ is used as the dynamic threshold $\epsilon$, right? It's not clearly stated in the text.

---

### Official Review · Reviewer_p17i · 2024-11-02

**Soundness:** 3
**Presentation:** 3
**Contribution:** 3
**Rating:** 8
**Confidence:** 4

**Summary:**

The paper introduces a new approach to maintaining plasticity for deep reinforcement learning methods based on intuitions about cortical cortex expansion in cognitive science. The approach includes three components: 1) neuron regeneration, 2) dormant neuron pruning, and experience review. While neuron regeneration and dormant neuron pruning parts help maintain plasticity, the experience review reduces instability due to high plasticity. The authors test the effectiveness of their approach and its components in various RL environments and compare it against other baselines.

**Strengths:**

The authors present a novel approach that improves plasticity for deep reinforcement learning methods. The approach seems effective and achieves better performance than many existing methods, such as layer normalization, ReDo, and plasticity injection in many environments. The authors provided an extensive experimental study of their method in different environments (MuJoCO Gym and DMC) and with different learning algorithms (DrQ and TD3).

**Weaknesses:**

- The paper significantly lacks mathematical rigor. Here are some examples that are representative of these inaccuracies, although they don’t constitute an exhaustive list:

  - It should be $\breve{\theta} \subset \theta$ not $\breve{\theta} \in \theta$. Or more precisely, $\breve{\theta}_l \subset \theta_l, \forall l \in \\{1,...,N\\}$, where $N$ is the number of layers in the network.

  - In line 213, $\mathbb{I_{grow}}$ is not defined well. It should be a list, but you assign it with two random quantities added together so it looks like a vector or a scaler instead of a set. Additionally, how is the random function defined? The random function should output a set, which you then need to union with the other set, $\mathbb{I_{grow}}= RandomSet1 \cup RandomSet2 $ not $\mathbb{I_{grow}}= RandomSet1 + RandomSet2$. A complete, rigorous mathematical description is expected.
  - It should be $ArgTopK$ not $TopK$ in equation 2 and line 256
  - what does this mean to write $\mathbb{I_{prune}}= f(\theta_i) \leq 0$? The left side should be a list, and the right-hand side should be an inequality. It should be something like $\mathbb{I_{prune}} = \\{\texttt{index}(\theta_i) | f(\theta_i) \leq 0 \\}$.
  - what does it mean to have a dormant ratio of negative in line 301? The ratio possible values are in $[0,1]$.
- The paper presentation and writing are not clear.
  - The authors claim NE maintains a high level of elastic neurons (see line 60), but no definition of what elastic neuron is given. Is elasticity here something different from plasticity? How do we measure either of them?
  - The term plasticity is loosely used to represent activated neuron ratio (e.g., Figure 6f). A clear definition of what plasticity means should be presented. If plasticity is the activated neuron ratio, then the paper's approach does not actually address the loss of plasticity as claimed since in all figures where the activated neuron ratio is presented, we see a decrease in their percentage with the paper's approach, similar to other methods.
  - The algorithm is not complete. For example, the cosine annealing scheduler is missing, and the experience review part is not clearly shown. Additionally, Algorithm 1 works on the weight level, but the description from the text talks about neuron-level regeneration and pruning. The algorithm needs to reflect that.
  - Since the authors depend on the sparse network training framework as part of their approach. They should fully explain what the sparse network training framework is in writing and in the algorithm.
  - The process of experience review is not clear. The fluctuation rate of dormant neurons $\nabla f$ is a function of each unit, but the authors talk about some aggregate quantity. Is that new quantity a summation of all units in the network, $\nabla f = \sum_i f_i$? Why isn’t this part of the algorithm?
  - Since the authors chose to rely on the activated neuron ratio (1-dormant neuron ratio), equation 1 needs to reflect that, currently, the definition is mentioned in-text, whereas it should be highlighted in equation 1 instead of dormant neuron ratio, which the authors do not really use.
- Some claims in the paper are not supported by evidence.
  - The paper overclaims what their approach can address. The authors mention that their approach mitigates loss of plasticity primacy bias, reduces catastrophic forgetting, and strikes a stability-plasticity balance. Most of these claims are not supported by evidence. Using those terms loosely without being precise about what is being studied in an experiment makes the paper hard to navigate.
  - For example: “topology growth can effectively alleviate neuron deactivation and thus maintain the ability of policy learning to mitigate the loss of plasticity and alleviate the primacy bias.”--- It's unclear how the experiment shows loss of plasticity or primacy bias mitigation. The authors should instead only claim that their approach reduces the dormant neuron ratio and not claim anything about loss of plasticity or primacy bias.
  - The current ablation is not sufficient. Ideally, the authors should remove each component of the system: 1) neuron regeneration, 2) experience review, and 3) dormant neuron pruning. The authors did 1 and 2 but not 3. We need to know what happens if we remove dormant neuron pruning.
- Issues in empirical evaluation:
  - Many figures do not have labels on the axes, so it is hard to know (even after careful investigation) what is being varied. For example, the x-axis in Figure 5 has no label, and I don’t know what 0 to 3 means here. Other examples include but are not limited to Figure 2 (missing x-axis label), Figure 4 (what is the score in the y-axis), and Figure 6 (missing y-axis label).
  - The results are not statistically significant. A very low number of independent runs (7 runs) are used, and they have overlapping error bars in most figures. More independent runs are needed, especially since the error bars are overlapping. I suggest the authors run each algorithm for 30 independent runs in all of their experiments.
  - In section 5.2, a fixed number of episodes is used in each task, whereas a fixed number of steps should be used to have consistent amount of experience in each task.

**Minor issues:**

- The author defines the gradient as $L_t$. Then the sentence after that says it’s $\nabla L_t$.
- The name of the approach is not very representative of what the algorithm does. It’s called neuroplastic expansion, emphasizing the expansion part. A better name, such as neuroplastic regeneration and pruning, can be more representative and accurate.


&nbsp;
&nbsp;
&nbsp;

Overall, I believe this paper could serve as a good algorithmic contribution to the community if the authors addressed my concerns based on this feedback. So, I’m willing to increase my score given that 1) the authors tuned down the claims and made them modest such that they accurately reflect what is being studied by their experiments, 2) the authors fixed all mathematical inaccuracies and provided a completed algorithm, 3) the terminologies are used carefully precisely instead of loosely, and 4) the empirical work is improved through more independent runs and improved figures.
\
\
\
\
**Update:** The authors worked hard to address my concerns. It was a fruitful back-and-forth discussion that improved the paper's quality immensely. Since my concerns have been addressed, I'm happy to recommend acceptance.

**Questions:**

1. Why is it called “Elastic” Neuron Generation? What is elastic exactly here?
2. I’m not sure how experience review reduces learning instability. I’m not convinced by the revisiting-is-useful argument. Without experience review, sampling is iid, so temporally old samples are still revisited. Why is revisiting old samples with higher probability useful, particularly when the dormant neuron ratio is high?
3. To get the gradients to decide which weight to regenerate, you need to use the fully expanded network and backpropagate everything, then find the top k weights that do not exist in the actual network. Is that correct? If so, then this metric is inaccurate because it adds all weights that take part in backpropagation. The accurate way is to add one weight at a time and backpropagate gradients each time. This is, of course, very expensive since you need to have $N$ additional forward and backward passes. In contrast, the process you described makes some approximations that are not clearly presented.
4. Are both pruning and regeneration neuron/unit-based?
5. In section 5.2, the environments have different action spaces; how did you handle that?
6. The authors stated that resetting was deemed the most effective approach. But no references are given (line 467).
7. In Figure 8, why are there spikes in the activated neurons ratio?
8. What is meant by removing the difference in fitting ability in lines 217-218?

---

### Official Review · Reviewer_LFqg · 2024-11-12

**Soundness:** 2
**Presentation:** 3
**Contribution:** 2
**Rating:** 6
**Confidence:** 3

**Summary:**

This paper introduces a novel idea, Neuroplastic Expansion (NE), to address the problem of plasticity loss in reinforcement learning (RL). The paper is well-written and presents the concept clearly. However, there are some concerns, particularly regarding its contribution relative to existing work. If these concerns can be resolved, I would consider improving the rating from 5 to 6.

**Strengths:**

1.The paper is well-written.

2.The concept of Neuroplastic Expansion (NE) is well-motivated.

**Weaknesses:**

See questions.

**Questions:**

1. Previous work, such as Plasticity Injection[1], has already proposed dynamically growing an agent's network, please provide a detailed comparison.

2.Dynamically expanding the size of a neural network could potentially lead to policy instability. For instance, the policy before and after expansion might be inconsistent. However, the results reported in Figure 6 appear very stable. Please provide specific analyses and ablation studies demonstrating how NE maintains policy stability during network expansion.

3.It would be helpful if the authors could provide experiments or analyses to explain the impact of dead neurons during training. Do these neurons store explored knowledge that contributes positively to the learning process, or do they have a negative effect on training? Please provide some analyses and visualizations to illustrate their impact on the learning process.

[1]Deep Reinforcement Learning with Plasticity Injection.

---

### Meta-Review · Area_Chair_xHTb · 2024-12-21

**Metareview:**

This paper addresses the stability-plasticity dilemma inspired by dormant neuron pruning and expansion of connection topology inspired by cortical expansion in cognitive science. The approach is intuitive, but the actual details of expansion seem quite complicated. An additional technique used by this paper is called experience review for neuron consolidation, where basically experience replay from the initial quarter of the buffer is conducted if plasticity gain is bottlenecked. Other times, the experience replay goes on normally.

Careful analysis is conducted to establish that all three together can strike a superior balance between stability and plasticity, providing highly performant learning in Mujoco Gym and DM Control Suite environments.

The writing remains unclear, as the reviewers mentioned, making it difficult to understand what the algorithm exactly does (I appreciate the pseudocode) and what the experiments exactly are.

I recommend acceptance with caution and strongly advise the authors to undertake a comprehensive revision of the paper before final submission to address the clarity issues.

**Additional Comments On Reviewer Discussion:**

The reviewers are generally appreciative of the paper, and extensive discussion helped reach an agreement of opinion between the authors and the reviewers. Along the way, the paper also improved.

---

### Decision · Program_Chairs · 2025-01-22

Accept (Poster)